# An Evaluation of the OLM *Pneum*ID Real-Time Polymerase Chain Reaction to Aid in the Diagnosis of *Pneumocystis* Pneumonia

**DOI:** 10.3390/jof9111106

**Published:** 2023-11-15

**Authors:** Jessica S. Price, Melissa Fallon, Raquel Posso, Matthijs Backx, P. Lewis White

**Affiliations:** Public Health Wales Mycology Reference Laboratory, PHW Microbiology Cardiff, University Hospital of Wales, Heath Park, Cardiff CF14 4XW, UK; jessica.price2@wales.nhs.uk (J.S.P.); melissa.fallon2@wales.nhs.uk (M.F.); raquel.posso@wales.nhs.uk (R.P.); matthijs.backx2@wales.nhs.uk (M.B.)

**Keywords:** OLM PneumID, *Pneumocystis jirovecii*, PcP PCR, Pneumocystosis

## Abstract

Background: The use of the PCR to aid in the diagnosis of *Pneumocystis* pneumonia (PcP) has demonstrated excellent clinical performance, as evidenced through various systematic reviews and meta-analyses, yet there are concerns over the interpretation of positive results due to the potential presence of *Pneumocystis* colonization of the airways. While this can be overcome by applying designated positivity thresholds to PCR testing, the shear number of assays described limits the development of a universal threshold. Commercial assays provide the opportunity to overcome this problem, provided satisfactory performance is determined through large-scale, multi-centre evaluations. Methods: Retrospective case/control and consecutive cohort performance evaluations of the OLM *Pneum*ID real-time PCR assay were performed on DNA eluates from a range of samples sent from patients where “in-house” PCR had been performed as part of routine diagnostic testing. The clinical performance of the *Pneum*ID assay was determined before including it in a diagnostic algorithm to provide the probability of PcP (dependent on diagnostic evidence). Results: After being used to test 317 patients (32 with PcP), the overall performance of the *Pneum*ID assay was found to be excellent (Sensitivity/Specificity: 96.9%/95.1%). False positivity could be removed by applying a threshold specific to sample type (<33.1 cycles for BAL fluid; <37.0 cycles for throat swabs), whereas considering any positive respiratory samples as significant generated 100% sensitivity, making absolute negativity sufficient to exclude PcP. Incorporating the *Pneum*ID assay into diagnostic algorithms alongside (1-3)-β-D-Glucan testing provided high probabilities of PcP (up to 85.2%) when both were positive and very low probabilities (<1%) when both were negative. Conclusions: The OLM *Pneum*ID qPCR provides a commercial option for the accurate diagnosis of PcP, generating excellent sensitivity and specificity, particularly when testing respiratory specimens. The combination of PcP PCR with serum (1-3)-β-D-Glucan provides excellent clinical utility for diagnosing PcP.

## 1. Introduction

The use of the PCR to aid in the diagnosis of *Pneumocystis* pneumonia (PcP) has been in place for many years, yet concerns over how to best diagnose PcP remain (https://blogs.jwatch.org/hiv-id-observations/index.php/chaos-in-the-diagnosis-of-pneumocystis-pneumonia/2022/12/19/#comments; accessed on 22 December 2022). Culture plays no role in the clinical diagnosis of PcP, with the histological examination of tissue and microscopic investigation of fluids providing a proven diagnosis. While PCR provides improved sensitivity over both conventional and immunofluorescence (IF) microscopy, PCR false positivity associated with the detection of airway colonization or as a result of contamination during the molecular process potentially undermine its clinical utility. However, systematic reviews and meta-analyses of the use of PcP PCR do not appear to reflect these concerns. Despite the obvious bias associated with using a reference standard (e.g., IF-microscopy) with lower sensitivity, the specificity of PcP PCR remains greater than 90%, even when testing less invasive upper respiratory tract specimens [1,2,3,4]. While the positive predictive value of a positive PcP PCR is less than 90%, the post-test probability is sufficient (66–82%) to warrant treatment in patients with suspected PcP. This is highlighted by the excellent positive likelihood ratios (9.9–16.2) and certainly reflects a marked increase over the pre-test probability (i.e., incidence) of disease [1,2,3]. The onset of quantitative PcP PCR, where higher fungal burdens can be determined, has the potential to further improve specificity. However, defining universal thresholds is not possible given the range of both locally developed (i.e., in-house) and commercial PcP PCR assays currently available [5]. Consequently, local validation and the clinical interpretation of results is required to develop positivity thresholds for differentiating between infection and colonization. For commercial assays, currently, between-centre validations are insufficient to confidently define thresholds that can be universally applied. The Fungal PCR initiative (FPCRI) is in the process of standardizing PcP PCR methodology, which should support the wider application of PcP PCR, which will be further enhanced by the availability of commercial assays [6,7].

One such commercial PcP PCR assay is the OLM *Pneum*ID, which generated high analytical sensitivity with quantification cycles (Cq) values within the top 5 of 19 assays evaluated in an FPCRI analytical study [7]. Results were rapid, within 2–3 h, when coupled with a range of automated nucleic acid extraction and qPCR platforms, and it is currently validated (CE-IVD) for respiratory wash and bronchoalveolar lavage (BAL) fluid. The assay includes an internal control that is extracted with all samples and monitors extraction efficiency and PCR inhibition but importantly also includes a PCR targeting the human β-globin gene, which has been shown to efficiently determine sample quality and avoid the reporting of false results [8]. In a small clinical evaluation testing BAL fluid from 50 mainly HIV-negative patients (82%), the *Pneum*ID assay generated an AUC of 0.988 through receiver operator characteristic (ROC) analysis, with a Cq threshold of 34.48, generating a sensitivity of 100% and specificity of 91.9%, while using an earlier (lower) Cq threshold of 26.68 cycles generated a sensitivity of 84.6% and specificity of 100% [9].

To provide further clinical validation of the *Pneum*ID assay, a retrospective case/control study was performed to provide sufficient cases of PcP. To avoid sample selection bias and also minimize the impact of storage degradation, a retrospective, consecutive cohort study was performed, where nucleic acid was stored for a minimal period (7–14 days) following primary PCR testing with the “in-house” assay.

## 2. Materials and Methods

### 2.1. Study Design

The initial study was a retrospective, anonymous case/control evaluation of performance of the *Pneum*ID qPCR assay using DNA extracted from clinical material (BAL fluid (n = 68), throat swabs (n = 56), serum (n = 11), induced sputum (n = 1)) previously tested using the in-house qPCR assay as a part of the routine fungal diagnostic work-up of BAL fluid (PcP PCR, *Aspergillus* PCR, Galactomannan EIA, fungal culture for non-*Pneumocystis* pathogens and microscopy) or from patients with suspected PcP (BAL, non-directed bronchial lavage (NBL), throat swab, induced sputum, serum/plasma) [10,11]. Samples for this part of the study were selected based on the previous “in-house” PcP PCR result to provide sufficient (>20) potential PcP cases, complemented with at least three-fold the number of control samples. Sample DNA was stored at −80 °C for a median of 18 days (range 3–576 days) for performance evaluation purposes prior to retesting with *Pneum*ID assay.

The retrospective case/control study was followed by a retrospective consecutive cohort evaluation of the PneumID qPCR assay, testing all samples that had been tested using the “in-house PCR” over a one-month period. For this study, the nucleic acid that was originally tested was stored at −20 °C, with PneumID testing performed within 7–14 days of primary “in-house” PCR testing. DNA was originally extracted from 77 BAL fluid samples, 66 throat swabs, 6 serum samples, 3 induced sputum samples, 2 NBL, and 1 lung swab.

Case definition was performed, blinded to the *Pneum*ID result, using the EORTC/MSGERC definitions to classify PcP, with a slight amendment, implemented to improve the probability of diagnosis [12]. Probable PcP was defined when a patient with host factors had radiology typical of PcP (e.g., bilateral ground glass opacification) and was positive by both “in-house” PcP PCR and serum (1-3)-β-D-Glucan (BDG). In addition, probable PcP was defined in the absence of BDG positivity (BDG negative or not performed) when PcP PCR was positive with a Cq below designated thresholds (<38 cycles and <36 cycles for upper and lower respiratory tract samples, respectively). These thresholds have previously been established as having optimal specificity and positive likelihood ratios for the “in-house” qPCR assay [11]. Patients with host factors and radiology typical of PcP with only BDG positivity that was not associated with an alternative fungal infection or only PcP PCR positivity but with a Cq later than the designated thresholds were considered as possible PcP. Patients with host factors and radiology typical of PcP that could not be attributed to another infection but lacking supporting mycology were classified as suspected PcP. Mycological positivity in the absence of typical radiology or clinical presentation of PcP were considered false positives, and were classified as controls, along with all patients lacking any clinical, radiological, and mycological evidence typical of PcP.

Data obtained to aid in the clinical interpretation of the routine diagnostic PCR and BDG results were retrospectively collated as an anonymous performance evaluation, with no impact on patient management. Following prior discussions with the local research board, this study was not considered to pertain to research under UK National Health Service guidance and therefore did not require ethical approval.

### 2.2. Nucleic Acid Extraction

For blood samples, total nucleic acid was extracted from 500 µL of serum/plasma on the Biomerieux eMAG extraction platform version 1.0.2 (Biomerieux UK Ltd., Basingstoke, UK) using the Generic_3.0.4 extraction protocol with the nucleic acid eluted in 75 µL. For respiratory samples, a respiratory swab or 200 µL of BAL, NBL, or induced sputum was added to 900 µL of Nuclisens lysis buffer mixed with a vortex and left to digest for 10 min, after which 200 µL of this buffer was added to a 2.0 mL of Nuclisens lysis buffer for extraction as for blood. Both positive and negative extraction controls were included, together with an internal control DNA target (diluted *Neisseria meningitidis* DNA) spiked into every sample.

### 2.3. In-House Pneumocystis qPCR Amplification

The “in-house” PcP qPCR assay involved amplifying a 77 bp of the mitochondrial 26S rDNA multi-copy gene using 5.0 µL of template in final reaction volume of 20 µL for 45 cycles on the ABI 7500 real-time PCR platform (ThermoFisher, Basingstoke, UK) using oligonucleotides as previously described [10]. An internal control PCR targeting the *CtrA* gene of *N. meningitidis* and Human RNAse P gene was performed to monitor for individual sample nucleic acid extraction efficiency and PCR inhibition while also determining sample quality.

### 2.4. PneumID Real-Time PCR Amplification

The OLM *Pneum*ID assay targets the large mitochondrial subunit and uses 6 µL of DNA template in final reaction volume of 20 µL on the Qiagen Rotorgene Q 6.0 HRM instrument (Qiagen UK Ltd., Manchester, UK). The assay was run for 45 cycles, with a positivity threshold of 0.01 normalized fluorescent units. As this assay used previously extracted nucleic acid, the internal control target DNA specific to the *Pneum*ID assay was spiked into the PCR master mix to only monitor for PCR inhibition, with individual sample quality and extraction efficiency already assessed by the “in-house” PcP PCR assay.

### 2.5. (1-3)-β-D-Glucan Testing

The BDG concentration in serum was determined using the Fungitell Assay (Associates of Cape Cod, Liverpool, UK) testing 5 µL of serum in duplicate, according to manufacturer’s instructions, with a positivity threshold of 80 pg/mL. Samples with a BDG concentration of between 60 and 79 pg/mL were considered indeterminate, and samples below 60 pg/mL were considered negative.

### 2.6. Statistical Analysis

To determine the clinical accuracy of the *Pneum*ID assay, the positivity rate in samples originating from cases was compared to the false positivity rate in the control samples. Total, positive, and negative observed agreement between the “in-house” and *Pneum*ID PCR assays was calculated, together with the generation of *Kappa* statistics when feasible. Clinical performance (sensitivity, specificity, positive and negative likelihood ratios, and diagnostic odds ratio) was determined through the construction of 2 × 2 tables, with probable and possible PcP considered as true cases and suspected patients or patients with no evidence of PcP used as the control population. For the initial case–control study, predictive values were not calculated but were generated for the consecutive cohort evaluation. For the comparison of proportionate values, ninety-five percent confidence intervals and, when required, *p* values (Fisher’s exact test; *p* ≤ 0.05 considered significant) were generated. To determine the optimal Cq threshold for defining *Pneum*ID positivity, ROC curve analysis was performed. Classification and regression tree (CART) analysis was performed to develop a combined predictive algorithm for PcP involving BDG and *Pneum*ID testing. Statistical analyses were performed using Graphpad Prism 5 (Graphpad Software, La Jolla, CA, USA) and Microsoft Excel 2016.

## 3. Results

### 3.1. Retrospective Case/Control Study—Sample Positivity

A total of 136 samples from 105 patients (25 PcP cases (32 samples) and 80 controls (104 samples) were tested. Overall, 37 (27.2%, 95% CI: 20.4–35.2) of the 136 samples were considered positive by the *Pneum*ID assay, comparable with the “in-house” qPCR (36/136, 26.5%, 95% CI: 19.8–34.5). A total of 31 (96.9%, 95% CI: 84.3–99.5) of the 32 samples from PcP cases were *Pneum*ID-positive, and concordance with the “in-house” assay was 100%. Moreover, 6 (5.8%, 95% CI: 2.7–12.0) of the 104 control samples were falsely deemed positive by the *Pneum*ID assay, comparable with the “in-house” qPCR (5/104, 4.8%, 95% CI: 2.1–10.8). Four false positive results were positive by both PCR tests, generating an observed agreement of 97.1% (101/104, 95% CI: 91.9–99.0) between the assays when testing control samples. Overall agreement between the assays was 97.8% (133/136, 95% CI: 93.7–99.3), generating a *Kappa* of 0.943, indicating excellent agreement. The *Pneum*ID true positivity rate when testing case samples was significantly greater than the false positivity rate when testing controls (difference: 91.1%, 95% CI: 77.0–95.1, *p* < 0.0001).

A total of 14 (20.6%, 95% CI: 12.7–31.6) of the 68 BAL fluids were positive, according to the *Pneum*ID, including 11/11 (100%, 95% CI: 74.1–100) BAL fluids from PcP cases and 3/57 (5.3%, 95% CI: 1.8–14.4) control samples. A total of 15 (26.3, 95% CI: 16.7–39.0) of the 57 upper respiratory tract samples were deemed positive by the *Pneum*ID, including 13/13 (100%, 95% CI: 77.2–100) from PcP cases and 2/44 (4.6%, 95% CI: 12.6–15.1) from controls. Additionally, 8 (72.7, 95% CI: 43.4–90.3) of the 11 serum samples were deemed positive by the *Pneum*ID, including 7/8 (87.5%, 95% CI: 52.9–97.8) serums from PcP cases and 1/3 (33.3%, 95% CI: 6.2–79.2) from controls.

### 3.2. Retrospective Case/Control Study—Clinical Performance

The retrospective clinical performance of the *Pneum*ID when testing a range of specimens is shown in Table 1. Overall sensitivity was 100%, only dropping below this value when testing serum and the patient with a false negative serum sample was *Pneum*ID positive in the respiratory tract. Specificity, in general, was excellent (>90%), comparable when testing both upper and lower respiratory tract samples, and while it was numerically lower when testing serum, the number of control patients tested was minimal. Optimal diagnostic performance was achieved when any positive result (irrespective of Cq) was considered significant, generating a Youden’s statistic of 0.9423. However, ROC analysis demonstrated that *Pneum*ID-positive results with a Cq < 33.1 cycles were associated with 100% specificity but a reduced sensitivity of 64.0% (16/25, 95% CI: 44.5–79.8). The area under the ROC curve (AUC) was 0.9688 (95% CI: 0.9301–1.0). When testing BAL fluid, the AUC was 0.9888 (95% CI: 0.9701–1.0), and considering any positive result (irrespective of Cq) was optimal, generating a Youden’s statistic of 0.9474. *Pneum*ID-positive BAL fluids with a Cq < 33.1 cycles were associated with 100% specificity (Sensitivity: 88.9%) for a diagnosis of PcP. For upper respiratory tract samples, the AUC was 0.9895 (95% CI: 0.9705–1.0), and considering any positive result (irrespective of Cq) was optimal, generating a Youden’s statistic of 0.9545. *Pneum*ID-positive upper respiratory tract samples with a Cq < 36.7 cycles were associated with 100% specificity (Sensitivity: 75.0%) for a diagnosis of PcP. For serum/plasma samples, the AUC was 0.8750 (95% CI: 0.6606–1.0), and considering any positive result (irrespective of Cq) generated a Youden’s statistic of 0.5417. Optimal performance (Youden’s: 0.75) *Pneum*ID was associated with a Cq < 34.6 cycles, which was associated with 100% specificity (Sensitivity: 71.4%) for a diagnosis of PcP, but numbers, particularly control samples, were limited.

### 3.3. Retrospective, Consecutive Cohort Study—Sample Positivity

One hundred and fifty-five samples from 112 consecutive patients (seven PcP cases—eleven samples—and 105 control patients—one hundred and forty-four samples) were retrospectively screened via the *Pneum*ID assay. The true positivity rate for samples from cases was 81.8% (9/11, 95% CI: 52.3–94.9), compared to a false positivity rate for the control samples of 2.8% (4/144, 95% CI: 1.1–6.9) (difference 79.0%, 95% CI: 49.2–92.2, *p* < 0.0001). Sample concordance with the “in-house” qPCR assay was excellent (observed agreement: 95.5% (149/155; 95% CI: 91.0–97.8); Kappa statistic: 0.765). Agreement between the PCR assays for case-based samples was 90.9% (10/11, 95% CI: 62.3–98.4), compared to 96.5% (139/144, 95% CI: 92.1–98.5) for the control samples. All discordant results were associated with PCR results with a late Cq value (median Cq: 38.9 cycles).

A total of 5 (6.5%, 95% CI: 2.8–14.3) of 77 BAL fluids were deemed positive by the *Pneum*ID, including 2/2 (100%, 95% CI: 34.2–100) BAL fluids from PcP cases and 3/75 (4.0%, 95% CI: 1.4–11.1) control samples. A total of 5 (7.2, 95% CI: 3.1–15.9) of 69 upper respiratory tract samples were deemed positive by the *Pneum*ID, with all positive samples from PcP cases generating a true positivity rate of 83.3% (5/6 95% CI: 43.7–97.0). Two (33.3, 95% CI: 9.7–70.0) of six serum samples were deemed positive by the *Pneum*ID, with all positive samples being from PcP cases generating a true positivity rate of 66.7% (2/3, 95% CI: 20.8–93.9).

### 3.4. Retrospective, Consecutive Cohort Study—Clinical Performance

A total of 7 of the 112 patients were diagnosed with PcP, generating a pre-test probability of PcP (i.e., incidence) of 6.3% (95% CI: 3.1–12.3). Overall sensitivity and specificity across all sample types was 85.7% (6/7, 95% CI: 48.7–97.4) and 97.1% (102/105, 95% CI: 91.9–99.0), respectively (Table 2). Four PcP cases had respiratory swabs tested, three of which were *Pneum*ID-positive; three cases had serum tested, two of which were *Pneum*ID-positive (the case that was *Pneum*ID-negative in serum was also negative on the respiratory swab), and one case had only two BAL fluids tested, and both were *Pneum*ID-positive. In addition, 3 of 105 the control patients were *Pneum*ID-positive when the testing deep respiratory samples (BAL fluid in two patients and a lung swab in one patient). One false positive control patient was also falsely positive by the “in-house” qPCR when testing BAL fluid, and two further control patients were falsely positive by the “in-house” PCR but negative via *Pneum*ID when testing BAL fluid. No non-invasive samples (respiratory swabs, induced sputum, serum) were falsely positive via *Pneum*ID or “in-house” qPCR, and subsequently, the probability of PcP associated with a positive *Pneum*ID result was high (>98% for both respiratory swabs and serum). Across all sample types, the probability of PcP associated with a positive *Pneum*ID result was 67%, compared to 1.0% if the *Pneum*ID was negative.

ROC analysis identified an optimal threshold of 40 cycles (Youden’s statistic: 0.838), generating a sensitivity and specificity of 85.7% (95% CI: 48.7–97.4) and 98.1% (95% CI: 93.3–99.5), respectively. Applying a positivity threshold of <34 cycles generated 100% (95% CI: 96.5–100) specificity while providing 57.1% (95% CI: 25.1–84.2) sensitivity. The AUC was 0.9034 (95% CI: 0.7666–1.0). Given the limited number of PcP cases in the consecutive study, it was not possible to perform ROC analysis of the individual specimen types.

## 4. Discussion

The retrospective performance of the *Pneum*ID qPCR assay for the diagnosis of PcP is comparable across case/control and consecutive cohort studies when testing a range of clinical samples (Table 1 and Table 2). When both studies are combined, the overall sensitivity and specificity are excellent, permitting both the confirmation and exclusion of PcP (dependent on the result; Table 3). Performance when testing respiratory samples remains excellent, whether testing specimens sampled from the lower or upper airways. Performance when testing serum appears inferior, but sample numbers are limited and therefore disproportionately impacted by the limited number of erroneous results.

Previous evaluations of the *Pneum*ID are limited, with one published study testing BAL fluid [9]. The overall *Pneum*ID performance when testing BAL fluid appears comparable with the current evaluation (ROC AUC: 0.988 previous study vs. 0.991 current study—case/control and consecutive studies combined), although thresholds associated with optimal performance were lower in the previous study, potentially associated with differences in technical setups (e.g., setting of the fluorescent threshold) and differences in bronchoscopic sampling or the nucleic acid extraction protocol [9]. The authors of a previous study concluded it was not feasible to define an individual optimal Cq threshold due to the inevitable overlap between colonization and infection. However, they did define thresholds that attained 100% specificity or 100% sensitivity, which left a “grey-zone” that required further clinical or diagnostic interpretation. This dual threshold approach is not unusual in mycology and has been widely incorporated for the interpretation of galactomannan ELISA results when testing BAL fluid for the diagnosis of invasive aspergillosis [13]. Across the current combined studies and all sample types, a *Pneum*ID Cq value <33.1 cycles was associated with 100% specificity, whereas considering any positive result as significant provided the highest sensitivity (96.9%). The thresholds will likely vary according to sample type. For BAL fluids tested via *Pneum*ID across the entire current study, considering any positive result as significant provided 100% sensitivity, with 100% specificity obtained when positivity was associated with Cq values <33.1 cycles. For throat swabs, the ROC AUC across both studies was 0.970, with positive Cq values <37.0 cycles generating 100% specificity, with a maximum sensitivity of 93.8% being generated when considering any positive result as significant. Ideally, the breadth of any “grey-zone” separating optimal sensitivity and specificity will be minimal, but the interpretation of positive results within this zone can be aided by an understanding of the host/clinical risk of PcP, clinical presentation, and the availability of additional mycological evidence, such as serum BDG [14].

The performance of the *Pneum*ID assay did not appear to be influenced by the host’s HIV status. Across the two studies, 8/8 (100%, 95% CI: 67.6–100) of the HIV-positive patients with PcP were *Pneum*ID-positive, compared to 23/24 (95.8%, 95% CI: 79.8–99.3) of HIV-negative patients (difference: 4.2%, 95% CI: −20.2 to 28.5, *p*: 1.0). Incorporating the overall *Pneum*ID performance, as highlighted in Table 3, into diagnostic algorithms also involving BDG testing with performance adjusted according to HIV status, as previously described, shows that combining the OLM *Pneum*ID with the Associates of Cape Cod Fungitell assay provides diagnostic strategies that can both confidently exclude and diagnose PcP, irrespective of HIV status (Figure 1 and Figure 2) [15].

The post-test probability of PcP when both BDG and PCR are positive represents a significant increase over the pre-test probability, although the performance of this combined diagnostic strategy for confirming a diagnosis of PcP appears superior in the HIV-positive patient (Figure 2).

However, the algorithms currently described incorporate general positivity thresholds not necessarily associated with optimal specificity for confirming infection. Obviously, *Pneum*ID Cq thresholds associated with 100% specificity and subsequent 100% post-test probability have been identified, but applying these can significantly negate assay sensitivity. When combining both retrospective evaluations of the *Pneum*ID assay, the ROC AUC was 0.9550 (95% CI: 0.9095–1.0), and applying a positivity threshold of 36.7 cycles generated an improved specificity of 97.8% (181/184; 95% CI: 94.6–99.2) while maintaining a sensitivity of 84.4 (27/32; 95% CI: 68.3–93.1). Incorporating these parameters into the HIV-negative PcP algorithm provided a post-test probability of PcP of 45.1% when associated with a positive PCR result, while maintaining an exceptional low post-test probability of not having PcP (0.3%) when the PCR result was negative or positive beyond 36.7 cycles. While the post-test probability of PcP in the HIV-negative patient associated with a *Pneum*ID-positive (Cq < 36.7) remains less than 50%, it reflects a >21-fold increase over the pre-test probability and combining PCR with BDG (performance and thresholds as currently stated in Figure 1) generates a post-test probability of PcP of 80.6% when both tests are positive and 0.06% when both tests are negative.

An individual serum BDG test will unlikely provide sufficient sensitivity to confidently exclude PcP in the HIV-negative patient when negative [15,18]. Across this study, 28/32 (87.5%, 95% CI: 71.9–95.0) PcP cases were BDG-positive in serum, compared to 17/185 (9.2%, 95% CI: 5.5–14.2) of control patients (*p* < 0.0001), and at a PcP incidence of 2.1%, the probability of PcP in the HIV-negative patient with an isolated serum BDG-positive is only 16.1%. In this study, the median concentration for positive BDG samples from PcP cases and controls was >500 pg/mL (range: 149–500 pg/mL) to 197 pg/mL (range: 83–500 pg/mL), respectively (*p*: 0.003). Subsequently, further confidence in a diagnosis of PcP can be achieved when higher serum BDG concentrations are detected, with values >156 pg/mL associated with 100% specificity for the diagnosis of PcP in the HIV-negative patient [18]. Across the current study, the BDG concentration was >156 pg/mL in 16/24 HIV-negative PcP cases (Sensitivity: 66.7%, 95% CI: 46.7–82.0), compared to 11/182 of HIV-negative control patients (Specificity: 94.0%, 95% CI: 89.5–96.6). Incorporating this BDG positivity threshold alongside a *Pneum*ID Cq threshold of <36.7 cycles generates a post-test probability of PcP of 90.0% when both tests are positive, compared to a post-test probability of not having PcP of 0.1% when both tests are negative. In a previous study of the HIV population, a BDG threshold of 300 pg/mL generated sensitivity and specificity of 91% and 92% for the diagnosis of PcP, respectively [19]. In the current study, 7/8 HIV-positive PcP cases had a serum BDG concentration above this threshold, but none of the control HIV-positive patients.

Interestingly, one intensive care patient with a history of myasthenia gravis receiving prolonged corticosteroids (prednisolone 50 mg/day) presented with shortness of breath and general malaise and was both *Pneum*ID-positive (Cq: 34.7 cycles) and BDG-positive (>500 pg/mL) in serum. The patient responded to level 3 interventions on the ICU prior to the availability of the qPCR result and was transferred to a general ward without receiving anti-PcP or any antimicrobial therapy prior to hospital discharge, with a diagnosis of an exacerbation of myasthenia gravis. Upon the availability of the routine “in-house” qPCR result (Cq: 38 cycles), a chest radiograph was performed (both chest X-ray and chest CT), but no evidence typical of PcP was identified; however, given the mycological evidence and risk of PcP associated with continued corticosteroid use, the patient was prescribed PcP prophylaxis. A routine investigation into these potentially false positive mycology results highlighted that the patient had received intravenous immunoglobulin (IVIG), a known source of BDG false positivity, prior to BDG testing [20]. This highlights the need for the clinical interpretation of all results and that even when the post-test probability of an infection is high (Figure 1 and Figure 2), the diagnosis is rarely definitive (i.e., 100% certain). This is a diagnostic issue regularly encountered in mycology, where a proven diagnosis of infection is limited (e.g., culture/histology from a sterile site) and most cases of fungal disease are defined as probable IFD [12]. Cases of dual IFD can also occur in high-risk patients, occurring in the retrospective consecutive cohort, as two of the seven PcP cases likely had dual infection. In one HIV-positive patient PcP PCR, *Aspergillus* PCR and galactomannan EIA (I = 3.0) were positive in BAL fluid, and chest CT reported evidence typical of sub-acute aspergillosis in one lobe but also bilateral ground glass opacification, leading to a diagnosis of probable PcP/aspergillosis; the patient was successfully treated with voriconazole and Septrin. A second haematology patient being treated (voriconazole/ambisome) for possible IFD due to a chest CT indicating the presence of nodules/halos and a thick-walled cavity, despite *Aspergillus* PCR and galactomannan EIA negativity, became PcP PCR-positive during the course of initial IFD treatment and was successfully treated with septrin.

Of the eight remaining false positive *Pneum*ID patients, only one was BDG-positive, with a concentration >500 pg/mL; this patient was also deemed positive by the “in-house” qPCR when testing the BAL fluid. A further three patients were falsely positive in both qPCR tests when testing BAL fluids (n = 2) and a throat swab. Six of the nine false positive results were associated with the testing of deep respiratory samples (BAL fluid n = 5 and a lung swab). In summary, 2/9 patients with false positive *Pneum*ID results were also false positive for BDG, compared to 5/9 who were also deemed positive via “in-house” qPCR. Across both phases of the study, only one case of PcP was false negative by the *Pneum*ID assay when testing a throat swab and a serum. The solid cancer patient presented with worsening shortness of breath and was only weakly positive (Cq: 39.1) based on “in-house” qPCR when testing a throat swab but had a serum BDG > 500 pg/mL, chest radiology consistent with PcP, and responded to anti-PcP therapy.

The limitations of the present study include its retrospective nature, but it was hoped that the retrospective consecutive phase would minimize any sample selection bias that may have occurred during the case/control phase, although the consecutive arm only contains limited cases, an anticipated issue when studying diseases of relatively low incidence. It is hoped that by combining the two phases, the potential individual study limitations would be offset by the strength of the other phase. The study also lacks any cases of proven PcP, defined through histopathological or microscopic evidence of *Pneumocystis*. However, this approach provides an imperfect reference, weighted towards specificity but lacking sensitivity. The PcP cases defined in this study were based on consensus definitions but designed to provide enhanced specificity through the requirement for two mycological criteria (PcP PCR and BDG positivity) or a single mycological criterion associated with a significant fungal burden.

## 5. Conclusions

The OLM *Pneum*ID qPCR may be a commercial option for the accurate diagnosis of PcP, generating excellent sensitivity and specificity, particularly when testing respiratory specimens. When testing BAL fluids, negativity is useful for excluding PcP, whereas high burdens (<33.1 cycles) are associated with a high specificity for confirming infection, as is positivity (<37.0 cycles) when testing throat swabs. The combination of PcP PCR with serum BDG can enhance the probability of PcP when both tests are positive and excludes disease when both tests are negative, and diagnostic algorithms can be optimized through defined positivity thresholds. The clinical interpretation of mycology results remains paramount, even when both BDG and *Pneum*ID are positive, but particularly when results are discordant. The agreement between the PcP PCR tests is excellent (96.9%, 95% CI: 94.2–98.4), indicating consistent clinical performance despite the technical, designative, and interpretative differences and discordant results are associated with late Cq values and typically not associated with disease.

## Figures and Tables

**Figure 1 jof-09-01106-f001:**
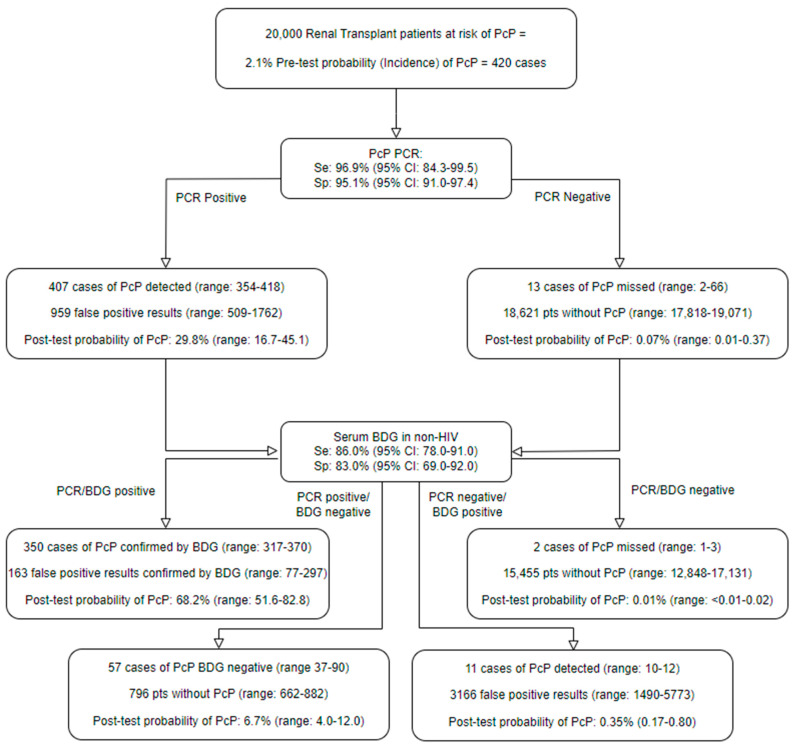
A diagnostic algorithm incorporating the OLM *Pneum*ID and the Associates of Cape Cod Fungitell (1-3)-β-D-Glucan (BDG) assay highlighting the probability of *Pneumocystis* pneumonia (PcP) in the HIV-negative patient. PcP PCR performance is based on the combined *Pneum*ID performance, as described in Table 3, with PCR positivity defined irrespective of the number of quantification cycles. BDG performance is based on the meta-analysis of Del Corpo et al., where the performance of BDG dependent on HIV status was evaluated, and any result with a BDG concentration >80 pg/mL was considered positive [15]. The incidence of PcP in the renal transplant population was derived from the study of Lee et al. [16].

**Figure 2 jof-09-01106-f002:**
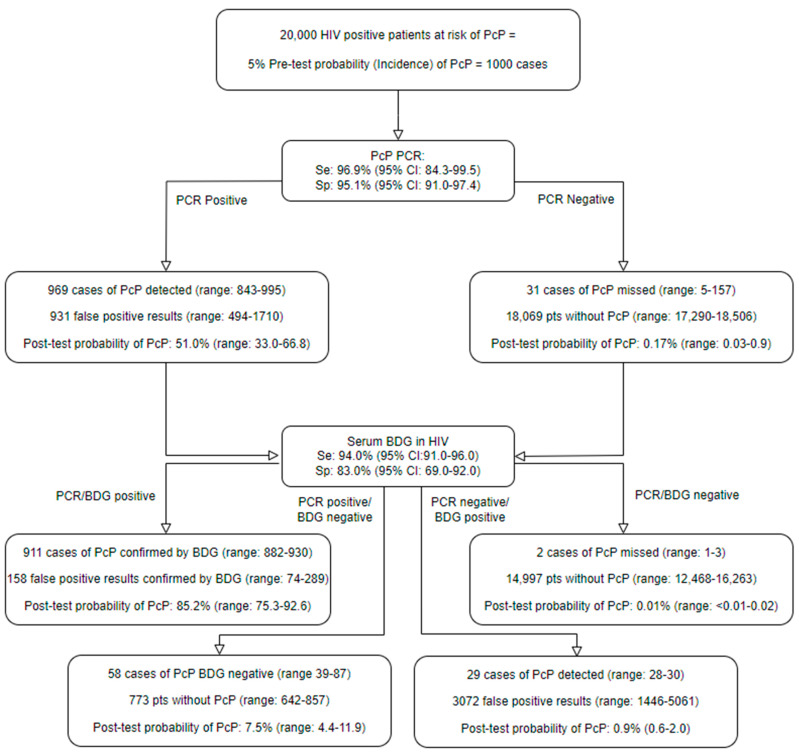
A diagnostic algorithm incorporating the OLM *Pneum*ID and the Associates of Cape Cod Fungitell (1-3)-β-D-Glucan (BDG) assay highlighting the probability of *Pneumocystis* pneumonia (PcP) in the HIV-positive patient. PcP PCR performance is based on the combined *Pneum*ID performance, as described in Table 3, with PCR positivity defined irrespective of the number of quantification cycles. BDG performance is based on the meta-analysis of Del Corpo et al., where the performance of BDG dependent on HIV status was evaluated, and any result with a BDG concentration >80 pg/mL was considered positive [15]. The incidence of PcP in the HIV cohort was derived from the study of Elango et al. [17].

**Table 1 jof-09-01106-t001:** Retrospective case/control performance of the OLM *Pneum*ID qPCR when testing various specimen types for the diagnosis of *Pneumocystis* pneumonia, with PCR positivity defined irrespective of the Cq value.

Population	Parameter				
	Se (%, 95% CI)	Sp (%, 95% CI)	LR + Tive	LR − Tive	DOR
All patients (n = 105 pts)	25/25 ^a^ (100, 86.7–100)	74/80 ^b^ (92.5, 84.6–96.5)	13.33	<0.001 ^c^	>13,330
BAL fluid (n = 51 pts)	9/9 (100, 70.1–100)	39/42 (92.9, 81.0–97.5)	14.08	<0.001 ^c^	>14,080
Upper respiratory tract (n = 50 pts) ^d^	12/12 (100, 75.8–100)	36/38 (94.7, 82.7–98.5)	18.87	<0.001 ^c^	>18,870
Serum (n = 10 pts)	6/7 ^e^ (85.7, 48.7–97.4)	2/3 (66.7, 20.8–93.9)	2.57	0.21	12.23

^a^ Three patients with PcP had two different sample types tested using PcP PCR (2× Serum/TS, 1× BAL/Serum). ^b^ Three patients without PcP had two different sample types tested using PcP PCR (3× BAL/TS). ^c^ Calculated using a sensitivity of 99.9% in order to provide a representative value in place of ∞. ^d^ Sample types: dry throat swabs (n = 49 pts) and induced sputum (n = 1 pt). ^e^ The one case patient who was PcP PCR-negative after testing serum was positive on two occasions after testing upper respiratory swabs. Key: Cq: Quantification cycle; BAL: Bronchoalveolar lavage; Se: Sensitivity. Sp: Specificity; LR + tive: Positive likelihood ratio; LR − tive: Negative likelihood ratio; DOR: Diagnostic odds ratio.

**Table 2 jof-09-01106-t002:** Retrospective performance of the OLM *Pneum*ID real-time PCR when testing various specimen types for the diagnosis of *Pneumocystis* pneumonia in a consecutive cohort population, with PCR positivity defined irrespective of the Cq value.

Population	Parameter						
	Se (%, 95% CI)	Sp (%, 95% CI)	PPV (%, 95% CI)	NPV (%, 95% CI)	LR + Tive	LR − Tive	DOR
All patients (n = 112 pts)	6/7 ^a^ (85.7, 48.7–97.4)	102/105 ^b^ (97.1, 91.9–99.0)	6/9 (66.7, 35.4–87.9)	102/103 (99.0, 94.7–99.8)	29.55	0.15	197
BAL fluid (n = 48 pts)	1/1 (100, 20.7–100)	45/47 (95.7, 85.8–98.8)	1/3 (33.3, 6.2–79.2)	45/45 (100, 92.1–100)	23.26	<0.001 ^f^	>23,260
Upper respiratory tract (n = 62 pts)	3/4 (75.0, 30.1–95.4)	58/58 ^c^ (100, 93.8–100)	3/3 (100, 43.9–100)	58/59 (98.3, 91.0–99.7)	>750 ^g^	0.25	>3000
Serum (n = 6 pts)	2/3 (66.7, 20.8–93.9)	3/3 (100, 43.9–100)	2/2 (100, 34.2–100)	3/4 (75.0, 30.1–95.4)	>667 ^g^	0.33	>2021
All deep respiratory samples (n = 50 pts)	1/1 ^d^ (100, 20.7–100)	46/49 ^e^ (93.9, 83.5–100)	1/4 (25.0, 4.6–69.9)	46/46 (100, 92.3–100)	16.39	<0.001 ^f^	>16,390

^a^ One patient with PcP had both a respiratory swab and serum tested via *Pneum*ID. ^b^ Two control patients had both a respiratory swab and BAL fluid tested via *Pneum*ID; one control patient had both a respiratory swab and serum tested via *Pneum*ID; one control patient had both a BAL fluid and a NBL fluid tested via *Pneum*ID, and one control patient had an induced sputum, a respiratory swab, and a BAL fluid tested via *Pneum*ID. ^c^ Includes 57 control patients where respiratory swabs were tested and 1 control patient where an induced sputum was tested via *Pneum*ID. One control patient where both a respiratory swab and an induced sputum were tested has only been included once. ^d^ Only includes one control patient where BAL fluid was tested via *Pneum*ID. ^e^ In addition to the 47 control patients where BAL fluid was tested using *Pneum*ID, this cohort includes 1 control patient where only NBL was tested and 1 control patient where a lung swab was tested via *Pneum*ID. One control patient where both NBL and BAL fluids were tested has only been included once. ^f^ Calculated using a sensitivity of 99.9% to provide a representative value in place of ∞. ^g^ Calculated using a specificity of 99.9% to provide a representative value in place of ∞. Key: Cq: Quantification cycle; Se: Sensitivity; Sp: Specificity; PPV: Positive predictive value; NPV: Negative predictive value; LR + tive: Positive likelihood ratio; LR − tive: Negative likelihood ratio; DOR: Diagnostic odds ratio; Pts: Patients; BAL: Bronchoalveolar lavage; NBL: Non-directed bronchial lavage.

**Table 3 jof-09-01106-t003:** Combined retrospective performance of the OLM *Pneum*ID real-time PCR when testing various specimen types for the diagnosis of *Pneumocystis* pneumonia, with PCR positivity defined irrespective of the Cq value.

Population	Parameter				
	Se (%, 95% CI)	Sp (%, 95% CI)	LR + Tive	LR − Tive	DOR
All patients (n = 217 pts)	31/32 ^a^ (96.9, 84.3–99.5)	176/185 ^a^ (95.1, 91.0–97.4)	19.78	0.03	659
BAL fluid (n = 99 pts)	10/10 (100, 72.3–100)	84/89 (94.4, 87.5–97.6)	17.86	<0.001 ^b^	>17,860
Upper respiratory tract (n = 112 pts)	15/16 (93.8, 71.7–98.9)	94/96 (97.9, 92.7–99.4)	44.67	0.06	745
Serum (n = 16 pts)	8/10 (80.0, 49.0–94.3)	5/6 (83.3, 43.7–97.0)	4.79	0.24	20
All deep respiratory samples (n = 101 pts)	10/10 (100, 72.3–100)	85/91 (93.4, 86.4–96.9)	15.15	<0.001 ^b^	>15,150

^a^ As outlined in Table 1 and Table 2, the cohort includes patients who have had more than one sample type tested via the *Pneum*ID assay. ^b^ Calculated using a sensitivity of 99.9% in order to provide a representative value in place of ∞. Key: Cq: Quantification cycle; BAL: Bronchoalveolar lavage; Se: Sensitivity; Sp: Specificity; LR + tive: Positive likelihood ratio; LR − tive: Negative likelihood ratio; DOR: Diagnostic odds ratio.

## Data Availability

Data are contained within the article.

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
