# Peer review of "An Evaluation of the OLM PneumID Real-Time Polymerase Chain Reaction to Aid in the Diagnosis of Pneumocystis Pneumonia"

_jof, 2023, doi:10.3390/jof9111106_

Round 1
Reviewer 1 Report
Comments and Suggestions for Authors
The study aims to evaluate the role of OLM PneumID real-time PCR assay in the diagnosis of PJP in immunocompromised HIV positive and HIV negative patients while evaluating sensitivity and specificity of this testing in respiratory specimens and serum.
The study is important and unique as the investigators address the challenge of attempting to diagnose and distinguish colonization from true infection in high-risk populations. They also add (1-3)-β-D-Glucan testing as part of the algorithm to improve testing performance.
The study addresses false positivity which has been a major barrier to clinical use by applying a positivity threshold specific to sample type (<33.1 cycles for BAL fluid; <37.0 cycles for throat swabs).
The retrospective design is associated with bias and limitations, but the investigators use of a retrospective case/control and consecutive cohort improves on decreasing limitations.
The investigators provide important compelling data on the performance of OLM PneumID qPCR for the accurate diagnosis of PJP while displaying the excellent sensitivity and specificity of this test mainly in respiratory specimens.
The references are appropriate. The tables and figures effectively display the data collected in the study. Well written and described study evaluating the role of PneumID in the diagnosis of PCP in immunocompromised patients.
Author Response
Reviewer 1
The study aims to evaluate the role of OLM PneumID real-time PCR assay in the diagnosis of PJP in immunocompromised HIV positive and HIV negative patients while evaluating sensitivity and specificity of this testing in respiratory specimens and serum.
The study is important and unique as the investigators address the challenge of attempting to diagnose and distinguish colonization from true infection in high-risk populations. They also add (1-3)-β-D-Glucan testing as part of the algorithm to improve testing performance.
The study addresses false positivity which has been a major barrier to clinical use by applying a positivity threshold specific to sample type (<33.1 cycles for BAL fluid; <37.0 cycles for throat swabs).
The retrospective design is associated with bias and limitations, but the investigators use of a retrospective case/control and consecutive cohort improves on decreasing limitations.
The investigators provide important compelling data on the performance of OLM PneumID qPCR for the accurate diagnosis of PJP while displaying the excellent sensitivity and specificity of this test mainly in respiratory specimens.
The references are appropriate. The tables and figures effectively display the data collected in the study. Well written and described study evaluating the role of PneumID in the diagnosis of PCP in immunocompromised patients.
Response: We thank the reviewer for taking the time to review our manuscript and for the positive comments returned.
Reviewer 2 Report
Comments and Suggestions for Authors
The manusccript presents a well-designed and structured study, although it is a bit complex to read due to the numerical part. As the authors mention, this study, due to its retrospective nature, has certain limitations.
My main criticism is that precisely the limitations of the study do not justify the conclusion, mainly in the following paragraph: “The OLM PneumID qPCR provides a commercial option for the accurate diagnosis of PcP, generating excellent sensitivity and specificity, particularly when testing respiratory specimens”. Since the study lacks cases of proven PcP, defined by histopathological or microscopic evidence of Pneumocystis, I consider that the most appropriate conclusion is: The OLM PneumID qPCR may be a commercial option for the accurate diagnosis of PcP, generating excellent sensitivity and specificity, particularly when testing respiratory specimens.
Minor corrections
Introduction
The authors should mention which region of the fungal genome is detected by PneumID qPCR.
Discussion
It would be interesting for the authors to discuss whether any of the samples positive by in-house PCR had a history of positivity to some other fungal agent, since cases of coinfection have been reported, for example with Histoplasma.
Author Response
Reviewer 2
he manusccript presents a well-designed and structured study, although it is a bit complex to read due to the numerical part. As the authors mention, this study, due to its retrospective nature, has certain limitations.
My main criticism is that precisely the limitations of the study do not justify the conclusion, mainly in the following paragraph: “The OLM PneumID qPCR provides a commercial option for the accurate diagnosis of PcP, generating excellent sensitivity and specificity, particularly when testing respiratory specimens”. Since the study lacks cases of proven PcP, defined by histopathological or microscopic evidence of Pneumocystis, I consider that the most appropriate conclusion is: The OLM PneumID qPCR may be a commercial option for the accurate diagnosis of PcP, generating excellent sensitivity and specificity, particularly when testing respiratory specimens.
Response: We thank the reviewer for taking the time to review our manuscript and we accept this viewpoint and have adjusted the manuscript accordingly.
Minor corrections
Introduction
The authors should mention which region of the fungal genome is detected by PneumID qPCR.
Response: We have included the following text “The OLM PneumID assay targets the large mitochondrial subunit and uses 6 µl of DNA template in final reaction volume of 20 µl on the Qiagen Rotorgene Q 6.0 HRM instrument.”
Discussion
It would be interesting for the authors to discuss whether any of the samples positive by in-house PCR had a history of positivity to some other fungal agent, since cases of coinfection have been reported, for example with Histoplasma.
Response: We agree this is an interesting discussion point, and we have this information to hand for consecutive cases and therefor included the following text in the discussion “Cases of dual IFD can also occur in high-risk patients and in the retrospective consecutive cohort, two of the seven PcP cases likely had dual infection. In one HIV positive patient PcP PCR, Aspergillus PCR and galactomannan EIA (I=3.0) were positive in BAL fluid and chest CT reported evidence typical of sub-acute aspergillosis in one lobe but also bilateral ground glass opacification, leading to diagnosis of probable PcP/aspergillosis; the patient was successfully treated with voriconazole and septrin. A second haematology patient being treated (voriconazole/ambisome) for possible IFD, due a chest CT indicating the presence of nodules/halos and a thick walled cavity, despite Aspergillus PCR and galactomannan EIA negativity became PcP PCR positive during the course of initial IFD treatment and was successfully treated with septrin.”
Reviewer 3 Report
Comments and Suggestions for Authors
Review for manuscript assigned as: jof-2659261 (Manuscript ID)
Dear Authors,
in my opinion your work is very interesting and has an exceptionally applied character. This paper contributes a lot to medical mycology, molecular diagnosis of Pneumocystis pneumonia (PcP) and molecular biology techniques. Study performed by Authors allow to show new possibilities in the diagnosis of pneumocystosis. The Authors put a lot of work into obtaining valuable results, preparing them meticulously and presenting them in a very accessible way. Nevertheless, some minor shortcomings could not be avoided, which is completely understandable.
All the figures and tables are appropriate for this type of article. In general, the paper has a logical flow. The abstract well correspond with the main aspects of the work and the literature is well selected for well prepared "Introduction" and "Discussion".
As a reviewer I am obligated to pay attention even to less important weak points of this work and all mentioned below comments should be carefully considered. All the comments below do not diminish my high assessment of this work.
Abstract
Page 1
,,When testing 317 patients...” sounds better instead of ,,When testing 317 pts...”
Page 1
The term ,,positivity" is used twice in the same sentence ,,False positivity could be removed by applying a positivity threshold specific to sample type...”
Page1
Each abbreviation, even within Abstract, should be expanded when first time used (in this case "BDG")
Introduction
What "in-house PCR" mean? Maybe, it will be better to explain for potential readers that "In-house" doesn't refer to a type of PCR, but it refers to the assay used.
In my opinion in Introduction it is worth mentioning at least briefly that PcP diagnostics based on cultivable methods is so far not possible and is currently based on molecular biology techniques and histological analysis for tissue examination.
Materials and Methods
2.1. Study design
The Authors write about ,,fungal culture" but to the best of my knowledge Pneumocystis jirovecii is not cultivable.
In the last sentence of the Introduction section, the Authors declare that DNA samples had been stored for a minimal period (7-14 days) but already in the "Materials and Methods" section the Authors write (quote) "Sample DNA was stored at -80° C for a median of 18 days (range 3-576 days), which is inconsistent.
The spelling "PCP PCR" or "PcP PCR" should be unified.
In the header 2.5 "(1-3)-β-". D-Glucan testing" dot should be deleted.
,,BDG concentration in serum was determined based on ...” sounds better and it is more appropriate than ,,Serum BDG testing...”
Results
Double bracket is used ,,(104 samples))”
Discussion
Table 2 and Table 3
In case of ,,All Deep respiratory” the term ,,Deep” should be written in lower case.
To the best of my knowledge, in healthy people the plasma concentration of (1-3)-β-D-glucan usually not exceed 10 pg/mL. The cut-off value defining a positive test result was established as 20 pg/mL, using the Fungitec assay. In my opinion it is worth to explain in the Discussion section why the Authors chose Fungitell assay among currently available Beta-D-Glucan Assays (reviewed in DOI: 10.1309/LM8BW8QNV7NZBROG). The cut-off value defining a positive test result for Fungitell was established as 60-80 pg/mL, which is significantly above the cut-off values adopted for other tests. Moreover, why are values above 80 pg/mL considered positive results instead of 60 pg/mL. I believe that it is worth explaining this issue in the Discussion section and dispelling the doubts of a potential reader.
Conclusions
As I suspect instead of ,,...even when both BDG and PneumID are positivity...” should be ,,...even when both BDG and PneumID are positive...”
,,designative differences” or ,,design differences” it is worth to think which one is more appropriate
Author Response
Reviewer 3
Review for manuscript assigned as: jof-2659261 (Manuscript ID)
Dear Authors,
in my opinion your work is very interesting and has an exceptionally applied character. This paper contributes a lot to medical mycology, molecular diagnosis of Pneumocystis pneumonia (PcP) and molecular biology techniques. Study performed by Authors allow to show new possibilities in the diagnosis of pneumocystosis. The Authors put a lot of work into obtaining valuable results, preparing them meticulously and presenting them in a very accessible way. Nevertheless, some minor shortcomings could not be avoided, which is completely understandable.
All the figures and tables are appropriate for this type of article. In general, the paper has a logical flow. The abstract well correspond with the main aspects of the work and the literature is well selected for well prepared "Introduction" and "Discussion".
As a reviewer I am obligated to pay attention even to less important weak points of this work and all mentioned below comments should be carefully considered. All the comments below do not diminish my high assessment of this work.
Abstract
Page 1
,,When testing 317 patients...” sounds better instead of ,,When testing 317 pts...”
Response: We thank the reviewer for taking the time to review our manuscript and we totally agree, apologies for this oversight!
Page 1
The term ,,positivity" is used twice in the same sentence ,,False positivity could be removed by applying a positivity threshold specific to sample type...”
Response: Sentence now reads “False positivity could be removed by applying a threshold specific to sample type (<33.1 cycles for BAL fluid; <37.0 cycles for throat swabs), whereas considering any positive respiratory samples as significant generated 100%, sensitivity, allowing absolute negativity sufficient to exclude PcP.”
Page1
Each abbreviation, even within Abstract, should be expanded when first time used (in this case "BDG")
Response: Agreed, Sentence now reads “Combination of PcP PCR with serum (1-3)-β-D-Glucan provides excellent clinical utility for diagnosing PcP”
Introduction
What "in-house PCR" mean? Maybe, it will be better to explain for potential readers that "In-house" doesn't refer to a type of PCR, but it refers to the assay used.
Response: We have defined the terminology for In-house PCR as follows, “However, defining universal thresholds is not possible given the range of both locally developed (i.e., in-house) and commercial PcP PCR assays currently available [5].”
In my opinion in Introduction it is worth mentioning at least briefly that PcP diagnostics based on cultivable methods is so far not possible and is currently based on molecular biology techniques and histological analysis for tissue examination.
Response: We have included the following statement “Culture plays no role in the clinical diagnosis of PcP, with histological examination of tissue and microscopic investigation of fluids providing a proven diagnosis.”
Materials and Methods
2.1. Study design
The Authors write about ,,fungal culture" but to the best of my knowledge Pneumocystis jirovecii is not cultivable.
Response: We have clarified this statement “…….part of the routine fungal diagnostic work-up of BAL fluid (PcP PCR, Aspergillus PCR, Galactomannan EIA, fungal culture for non-Pneumocystis pathogens and microscopy)”
In the last sentence of the Introduction section, the Authors declare that DNA samples had been stored for a minimal period (7-14 days) but already in the "Materials and Methods" section the Authors write (quote) "Sample DNA was stored at -80° C for a median of 18 days (range 3-576 days), which is inconsistent.
Response: The 7-14 days relates to the retrospective consecutive cohort study, whereas the median 18 days related to the retrospective case/control study. We feel this is already clearly stated in the text.
The spelling "PCP PCR" or "PcP PCR" should be unified.
Response: We search the document for PcP, and corrected any abbreviations from PCP to PcP.
In the header 2.5 "(1-3)-β-". D-Glucan testing" dot should be deleted.
Response: Good spot, this has been amended.
,,BDG concentration in serum was determined based on ...” sounds better and it is more appropriate than ,,Serum BDG testing...”
Response: text now reads “The BDG concentration in serum was determined using the Fungitell Assay…”.
Results
Double bracket is used ,,(104 samples))”
Response: Text corrected.
Discussion
Table 2 and Table 3
In case of ,,All Deep respiratory” the term ,,Deep” should be written in lower case.
Response: Text in tables corrected.
To the best of my knowledge, in healthy people the plasma concentration of (1-3)-β-D-glucan usually not exceed 10 pg/mL. The cut-off value defining a positive test result was established as 20 pg/mL, using the Fungitec assay. In my opinion it is worth to explain in the Discussion section why the Authors chose Fungitell assay among currently available Beta-D-Glucan Assays (reviewed in DOI: 10.1309/LM8BW8QNV7NZBROG). The cut-off value defining a positive test result for Fungitell was established as 60-80 pg/mL, which is significantly above the cut-off values adopted for other tests. Moreover, why are values above 80 pg/mL considered positive results instead of 60 pg/mL. I believe that it is worth explaining this issue in the Discussion section and dispelling the doubts of a potential reader.
Response: The kit was chosen as it is the most widely established BDG assay in the Western hemisphere, and we have used this assay for many years in routine service. These thresholds were developed by the kit manufacturer and they differ from other kits due to the different kinetics of reaction linked to the different sources of the main reagent. We used 80pg/ml rather than 60pg/ml, as 80pg/ml is the officially threshold for defining positivity. Given this manuscript is primarily focused on PcP PCR and word limits we do not think it is necessary to discuss these points.
Conclusions
As I suspect instead of ,,...even when both BDG and PneumID are positivity...” should be ,,...even when both BDG and PneumID are positive...”
Response: Good spot, this has been amended.
,,designative differences” or ,,design differences” it is worth to think which one is more appropriate
Response: Indeed, originally we meant design, but this likely covered by the technical differences, so we used designative differences to highlight differences in the how the assays denote positivity.